# Excitation and Tuning of Optical Tamm States in a Hybrid Structure with a Metal Film Adjacent to a Four-Layer Polymer–Liquid Crystal Stack

Victor Y. Reshetnyak [1,2] , Igor P. Pinkevych [1,*], Timothy J. Bunning [3] and Dean R. Evans [3]

1   Physics Faculty, Taras Shevchenko National University of Kyiv, 01601 Kyiv, Ukraine; victor.reshetnyak@gmail.com
2   School of Physics and Astronomy, University of Leeds, Leeds LS2 9JT, UK
3   Air Force Research Laboratory, Materials and Manufacturing Directorate, Wright-Patterson Air Force Base, Dayton, OH 45433, USA
*   Correspondence: ipinkevych@gmail.com

**Abstract:** Absorption, reflection, and transmission coefficients of the hybrid structure formed by a metal film and a holographic polymer–liquid crystal grating (HPLCG) are theoretically studied in the spectral region of the HPLCG band gap. HPLCG cells consist of four alternating layers, two layers of polymer and two layers of the same liquid crystal (LC), but with different orientations of the LC director. The appearance of reflection, transmission, and absorption peaks in the HPLCG band gap due to the excitation of optical Tamm states (OTSs) at the metal film–HPLCG interface is investigated. The dependence of the spectral manifestation of OTSs on the parameters of the hybrid structure is also studied. A comparison is made with the corresponding results for the case when HPLCG cells of a hybrid structure consist of one polymer layer and one LC layer (two-layer HPLCG).

**Keywords:** optical Tamm states; holographic polymer–liquid crystal grating; reflection coefficient





## 1. Introduction

In recent decades, intensive studies have been carried out on localized optical states arising at the interface between a metal film and distributed Bragg reflector (DBR) or two media with DBR properties. These states are called optical Tamm states (OTSs) [1–4] or Tamm plasmons if they can be associated with the excitation of plasmons in the metal film [5–10]. OTSs can be excited by an electromagnetic wave at any angle of incidence without an additional prism or grating and appear optically as narrow peaks/dips in the absorption/reflection spectra in the DBR band gap region. It is assumed that OTSs can become a good alternative to conventional surface plasmons in applications for sensors [11–14], optical switches, filters, emitters [15–21], and lasers [22,23].

Since liquid crystals (LCs) easily change their state under external fields, they are often used to tune Tamm plasmon-based devices [24–26]. In addition, since cholesteric LCs reflect light with circular polarization coinciding with the cholesteric LC helix [27], cholesteric LCs can be used instead of DBR in structures to excite OTSs [28–30]. LCs are also included in holographic polymer–liquid crystal gratings (HPLCGs). A HPLCG is a sequence of periodically alternating layers of polymer and nematic LCs with DBR properties. Such holographic gratings are obtained from the polymerization of a photosensitive mixture of monomers and LCs in the interference field of two intersecting coherent laser beams [31–38]. It was shown that HPLCGs can be used in a hybrid structure with a metal film to excite OTSs [39].

In the present paper, we theoretically study the excitation and control of OTSs in a hybrid structure composed of a metal film and an HPLCG consisting of a stack of four-layer cells. Each cell is formed by a sequence of polymer-LC1-polymer-LC2 layers, where LC1

and LC2 differ only in the director orientation. The HPLCG with four-layer cells make it possible to influence the OTSs over a wider frequency range of the HPLCG band gap than in the case when the HPLCG cells consist of only two layers. In addition, by changing the director orientation in the LC layers of the cell, it becomes possible to switch from OTSs in the band gap of the four-layer HPLCG to OTSs in a completely different spectral range corresponding to the band gap of the two-layer HPLCG, and vice versa.

The paper is organized as follows. Section 2 presents a theoretical model of a hybrid structure consisting of a metal film and a four-layer HPLCG, and derives analytical expressions for the reflection, transmission, and absorption coefficients in the HPLCG band gap. Results of numerical calculations and their discussion are presented in Section 3. Section 4 presents brief conclusions.

## 2. Materials and Methods

Let us consider a hybrid structure containing a metal film and a medium with DBR properties; for generality, we assume that there is an isotropic dielectric spacer between them (Figure 1a). The DBR medium is an HPLCG consisting of a repeated four-layer sequence polymer–LC1–polymer–LC2, where LC1 and LC2 differ only by the director direction (Figure 1b). Directing the Cartesian $z$-axis perpendicular to the layers of the hybrid structure, the HPLCG has a grating spacing $\Lambda$ along the $z$-axis and a thickness $L = N\Lambda$, where $N$ is an integer. Let $a$ and $d$ denote the thicknesses of the LC and polymer layers, respectively, such that $\Lambda = 2(a + d)$. The thicknesses of the metal film and the spacer are denoted as $l$ and $l_1$, respectively.

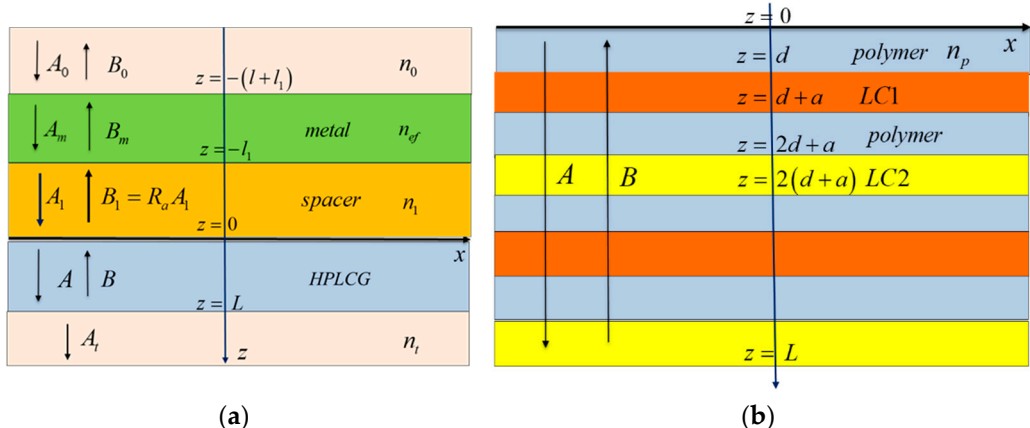

(a)                    (b)

**Figure 1.** (**a**) Schematic of the hybrid structure along with the directions of the propagating light beams. (**b**) Schematic of the 4-layer holographic polymer–liquid crystal grating (HPLCG) shown in (**a**). $n_0$, $n_{ef}$, $n_1$, $n_p$, and $n_t$ are the refractive indices of constituent materials defined in the text.

A monochromatic plane electromagnetic wave with amplitude $A_0$ and polarization along the $x$-axis is incident along the $z$-axis on the metal film and propagates through the structure, as shown in Figure 1a. The electric and magnetic field vectors of the wave in the isotropic medium in front of the metal film can be written as

$$\vec{E}(z,t) = E_0(z)\vec{e}_x e^{-i\omega t}, \ \vec{H}(z,t) = H_0(z)\vec{e}_y e^{-i\omega t},$$
$$E_0(z) = A_0 \exp(ikn_0 z) + B_0 \exp(-ikn_0 z),$$
$$H_0(z) = \frac{n_0}{\mu_0 c}[A_0 \exp(ikn_0 z) - B_0 \exp(-ikn_0 z)],$$

$$(1)$$

where $k = \omega/c$, $B_0$ is the reflected wave amplitude, and $n_0$ is the refractive index of the medium.

Denoting the complex refractive index of the metal as $n_{ef}$ and the refractive index of the dielectric spacer as $n_1$, the electric and magnetic field vectors of the wave in the metal film and the isotropic dielectric spacer can be written similarly to Equation (1).

One can write the boundary conditions for the electric and magnetic field vectors at $z = -(l + l_1)$ and $z = -l_1$ by taking into account that the amplitude of the wave reflected from the HPLCG, $B_1 = R_a A_1$, where $A_1$ is the amplitude of the wave incident on the HPLCG and $R_a$ is the amplitude of the reflection coefficient at the HPLCG boundary ($z = 0$). By solving the corresponding equations following from these boundary conditions, one can obtain an expression for the reflection coefficient of the hybrid structure in the following form:

$$R = \left| \frac{B_0}{A_0} \right|^2 = \left| \frac{\left( n_{ef} - n_0 \right) h_1 - \left( n_{ef} + n_0 \right) h_2}{\left( n_{ef} + n_0 \right) h_1 - \left( n_{ef} - n_0 \right) h_2} \right|^2, \tag{2}$$

$$\begin{aligned} h_1 &= \left( n_{ef} + n_1 \right) + \left( n_{ef} - n_1 \right) R_a \exp(i2kn_1 l_1), \\ h_2 &= \left[ \left( n_{ef} - n_1 \right) + \left( n_{ef} + n_1 \right) R_a \exp(i2kn_1 l_1) \right] \exp(i2kn_{ef} l), \end{aligned} \tag{3}$$

and the amplitude of the reflection coefficient, $R_a$, is still to be found.

To determine $R_a$, consider the electromagnetic field in the HPLCG region, $0 \leq z \leq L$. We assume the polymer layers of the HPLCG to be isotropic with the refractive index $n_p$, and the LC director can be reoriented in the $xz$-plane. It is convenient to present the director in the LC layers in the form $\vec{n}_i = (\cos \theta_i, 0, \sin \theta_i)$, $i = 1, 2$, where $\theta_i$ is the director angle with respect to the $x$-axis in the $i$-th layer of the cell. For simplicity, we neglect the director inhomogeneity in the LC layers and replace the angles $\theta_i$ with its average value $\bar{\theta}_i$, which is the same at all points of the LC layer.

Next, we will follow an approach like that used in [39]. Due to the LC anisotropy, the electric field vector of the electromagnetic wave propagating in the HPLCG has components $E_x$ and $E_z$ in each LC layer, where $E_z = -\left( \varepsilon_{zx}^{LC} / \varepsilon_{zz}^{LC} \right) E_x$ [40]. In a typical LC $E_z < 0.1\, E_x$; therefore, we can neglect the contribution from $E_z$ so the wave equation for the electric field in the HPLCG reduces to

$$\Delta E_x(z) + \frac{\omega^2}{c^2} \varepsilon_{xx} E_x(z) = 0, \tag{4}$$

where $\varepsilon_{xx}$ is the component of the HPLCG dielectric matrix, which in the case of the 4-layer HPLCG can be written as follows:

$$\hat{\varepsilon} = \begin{cases} n_p^2 \hat{I}, \; 0 < z < d \\ \hat{\varepsilon}_{LC1}, \; d < z < d + a \\ n_p^2 \hat{I}, \; d + a < z < 2d + a \\ \hat{\varepsilon}_{LC2}, \; 2d + a < z < 2(d + a) \end{cases}, \tag{5}$$

where $\hat{I}$ is the unit matrix and $\hat{\varepsilon}_{LC1}$ and $\hat{\varepsilon}_{LC2}$ are the dielectric matrixes of the first and second LC layers.

Consider the case when the incident beam wavelength is close to the longest wavelength, satisfying the Bragg resonance condition for HPLCG, $\lambda_r = 2\Lambda \bar{n}$, where $\bar{n}$ is the average refractive index of the HPLCG. Then, to solve Equation (5), one can use the coupled wave method [41] and obtain the Kogelnik equations [42] for the amplitudes $A$ and $B$ of electromagnetic waves in the HPLCG (see Figure 1a). Solving these equations, along with the boundary conditions at the HPLCG boundaries $z = 0$ and $z = L$, we obtain the amplitude of the reflection coefficient from the HPLCG, $R_a = B_1 / A_1$,

$$R_a = \frac{g_1 - g_2 w \exp(-2\gamma L)}{g_3 - g_4 w \exp(-2\gamma L)}, \tag{6}$$

where

$$w = \frac{(n_t - \bar{n} + \chi_{-1}/k) + (n_t + \bar{n} - \chi_1/k) r}{(n_t - \bar{n} + \chi_{-1}/k) + (n_t + \bar{n} - \chi_1/k) s}, \tag{7}$$

$$g_1 = (n_1 - \overline{n} + \chi_{-1}/k) + (n_1 + \overline{n} - \chi_1/k)r, \quad g_2 = (n_1 - \overline{n} + \chi_{-1}/k) + (n_1 + \overline{n} - \chi_1/k)s,$$
$$g_3 = (n_1 + \overline{n} - \chi_{-1}/k) + (n_1 - \overline{n} + \chi_1/k)r, \quad g_4 = (n_1 + \overline{n} - \chi_{-1}/k) + (n_1 - \overline{n} + \chi_1/k)s, \tag{8}$$

$$\overline{n} = \sqrt{dn_p^2/\Lambda + (1/2)(1 - d/\Lambda)\left[2n_o^2 + (n_e^2 - n_o^2)(\cos^2\overline{\theta}_1 + \cos^2\overline{\theta}_2)\right]},$$
$$\gamma \approx (\chi_1\chi_{-1} - \delta^2)^{1/2}, \quad r = i\chi_{-1}/(\gamma - i\delta), \quad s = \chi_{-1}/r\chi_1, \quad .\delta = k\overline{n} - \frac{\pi}{\Lambda} \tag{9}$$

In Equations (7)–(9), $\chi_{\pm1} = k\varepsilon_{\pm1}/2\overline{n}$ and $\varepsilon_{\pm1}$ are defined by the following expression:

$$\varepsilon_m = \frac{i}{2m\pi}$$
$$\times \left\{\left[n_p^2 - n_o^2 - (n_e^2 - n_o^2)\cos^2\overline{\theta}_1\right]\left(e^{-im\frac{2\pi}{\Lambda}d} - (-1)^m\right) + \left[n_p^2 - n_o^2 - (n_e^2 - n_o^2)\cos^2\overline{\theta}_2\right]\left(e^{im\frac{2\pi}{\Lambda}a} - 1\right)\right\}, \tag{10}$$

where $m = \pm1$, $n_o$ and $n_e$ are the LC refractive indices for the electromagnetic waves polarized perpendicular and parallel to the LC director, respectively.

Thus, Equations (2), (3), and (6)–(10) determine the reflection coefficient for the hybrid structure metal–HPLCG with four-layer cells: $R = |B_0/A_0|^2$.

Similarly, in using the boundary conditions at $z = L$, one can calculate an amplitude $A_t$ of the transmitted wave and obtain an expression for the transmission coefficient of the hybrid structure:

$$R_t = \left|\frac{A_t}{A_0}\right|^2 = \left|\frac{4n_0 n_{ef}(1 + R_a)}{\left(n_{ef} + n_0\right)h_1 - \left(n_{ef} - n_0\right)h_2}\frac{[1 + r - (1+s)w]\exp(ikn_{ef}l)\exp(-\gamma L)}{[1 + r - (1+s)w\exp(-2\gamma L)]}\right|^2, \tag{11}$$

and the absorption coefficient $A_{ab}$ of the hybrid structure can now be calculated as $A_{ab} = 1 - R - R_t$.

When the director angles $\overline{\theta}_1$ and $\overline{\theta}_2$ are equal, $\overline{\theta}_1 = \overline{\theta}_2 = 0$, the layers LC1 and LC2 become identical. In this case, we obtain the hybrid structure with the two-layer HPLCG and half the grating spacing discussed in [39]. The resulting expressions for the reflection, absorption, and transmission coefficients are determined by the same formulas as for the hybrid structure metal–HPLCG with the four-layer cells, but the term $\varepsilon_m$ in Equation (10) must be replaced by (see [39]) the following:

$$\varepsilon_m = i\frac{n_p^2 - n_o^2 - (n_e^2 - n_o^2)\cos^2\overline{\theta}}{2\pi m}[\exp(-im2\pi d/\Lambda) - 1], \tag{12}$$

where $\Lambda$ is the grating spacing of the two-layer HPLCG.

## 3. Results of Numerical Calculations and Discussion

For numerical calculations of the reflection, absorption, and transmission coefficients, Au and Ag metal films are considered with the frequency dependence of their complex refractive index $n_{ef}$ obtained in [43]. For the LC layers in the HPLCG, we consider LC 5CB and E7. The LC refractive indices and their frequency dispersion are taken from [44]. The refractive index of the polymer layers is assumed to be $n_p = 1.7$. The refractive indices of the medium before the metal film and after the HPLCG are $n_0 = 1$ and $n_t = 1$, respectively. The thicknesses of the LC and polymer layers are $a = 0.2\,\Lambda$ and $d = 0.3\,\Lambda$, respectively (see Figure 1b). The HPLCG grating spacing $\Lambda = 0.2\,\mu m$ is chosen such that the HPLCG band gap is located in the visible region of the spectrum. Other parameters of the system, i.e., thickness $l_1$ and refractive index $n_1$ of the spacer, thickness of the metal film $l$, number of cells in the HPLCG $N$, and director angles $\overline{\theta}_1$ and $\overline{\theta}_2$ in the LC1 layer and the LC2 layer, respectively, are variable.

Figure 2 shows the calculated spectral distribution of the reflection, absorption, and transmission coefficients in the HPLCG band gap region of the hybrid structure with a metal film of Au and LC 5CB, where the HPLCG thickness is $L = 70\,\Lambda$ (Figure 2a) and $L = 45\,\Lambda$ (Figure 2b). Figure 2a shows that the HPLCG thickness is substantially thick such

that the transmission is zero throughout the spectral region of the HPLCG band gap, but at a wavelength of 653 nm there is a peak in the absorption coefficient and a dip in the light reflectance coefficient. This dip in the reflectance occurs due to the light absorption, which is identified with the excitation of OTS at a wavelength of 653 nm [1–4]. When the HPLCG is not thick enough, some of the light energy at this wavelength passes, causing a transmittance peak, as seen in Figure 2b for HPLCG thickness $L = 45 \, \Lambda$.

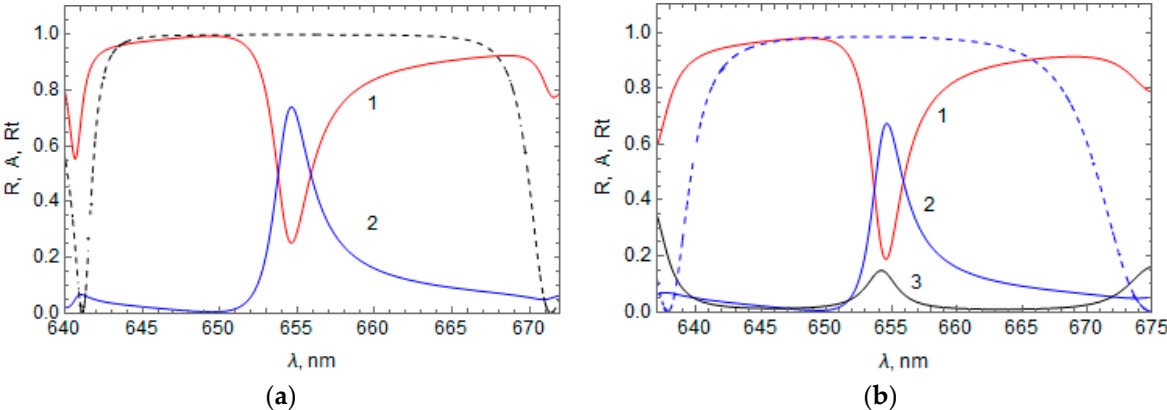

**Figure 2.** The reflection (1), absorption (2), and transmission (3) coefficients of the hybrid structure in the HPLCG band gap region (dotted line). The HPLCG thickness: (**a**) $L = 70 \, \Lambda$ and (**b**) $L = 45 \, \Lambda$. $\bar{\theta}_1 = \pi/2$, $\bar{\theta}_2 = 0$, $l = 25 \, \text{nm}$, $l_1 = 50 \, \text{nm}$. The structure comprises an Au layer and 5CB liquid crystal molecules. In Figure 2a, there is no light transmission.

Figure 3 shows the spatial profile of the electromagnetic field energy inside the hybrid structure at a wavelength corresponding to the reflection/absorption dip/peak shown in Figure 2a. The calculations were performed using the COMSOL Multiphysics, Electromagnetic Waves, Frequency Domain interface. The interface uses the finite element method to solve the frequency domain form of Maxwell's equations. There are two perfectly matched layers (PMLs) at the top and bottom of the modeling domain with scattering boundary conditions. We used periodic boundary conditions at the left and right sides of the domain. The electromagnetic wave is excited using a periodic port. The localization of the energy near the metal film and its decreasing oscillations at a distance comparable to the light wavelength is observed, which is typical for OTSs.

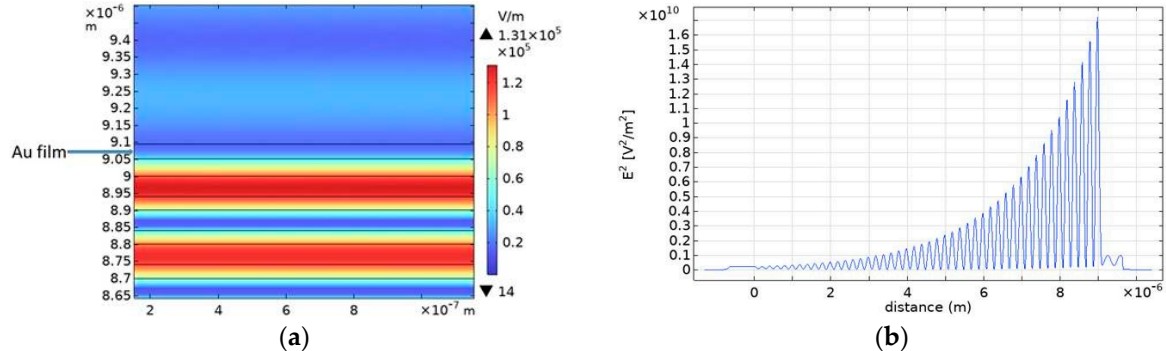

**Figure 3.** Spatial distribution of the square of the electric field strength corresponding to the reflection/absorption dip/peak shown in Figure 2a.

The spectral position and magnitude of the dip/peak connected with the OTS excitation depends on the type of metal and LC used in the hybrid system. We illustrate this in an example of the absorption coefficient for the hybrid structure with Au and Ag metal films and the same LC 5CB (Figure 4a) and for the hybrid structure with Au film and different LC, 5CB, and E7 (Figure 4b). Other parameters are the same for all cases.

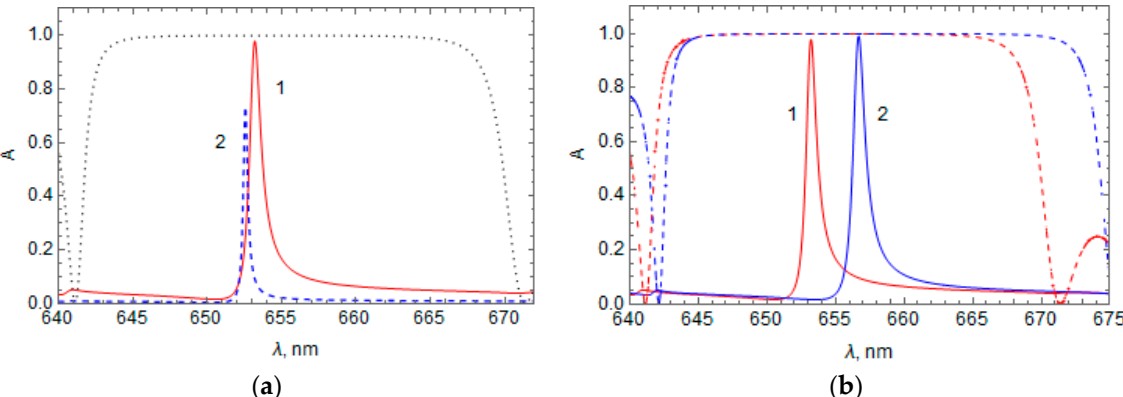

**Figure 4.** Dependence of the absorption coefficient peak on the type of the metal film and LC.
(**a**) Au + 5CB—(1), Ag + 5CB—(2); (**b**) Au + 5CB—(1), Au + E7—(2). $L = 70\,\Lambda$. $\bar{\theta}_1 = \pi/2$, $\bar{\theta}_2 = 0$,
$l = 45\,\text{nm}$, $l_1 = 50\,\text{nm}$.

Since the absorption and transmission peaks in the HPLCG band gap are obviously associated with the reflectance dip at the OTC wavelength, in what follows we will use only the reflectance data of the hybrid structure to illustrate the OTS spectral manifestations. The dependence of the reflection coefficient on the metal film thickness is shown in Figure 5a,b for the Au and Ag films, respectively. In both cases, the HPLCG with LC 5CB is considered. From Figure 5, it is seen that there is an optimal value of the metal film thickness, $l = 45\,\text{nm}$ for Au film and $l = 60\,\text{nm}$ for the Ag film, providing a maximum dip in the reflection coefficient at the OTS wavelength. Note that for a system with the two-layer HPLCG with the same grating spacing as in the case of the four-layer HPLCG (that is, in the region of the same wavelength), we do not observe the OTS wavelength dependence on the metal film thickness [39]. This indicates greater sensitivity of the four-layer system to the metal film thickness and, therefore, creates more opportunities to control the spectral position and depth of the reflection peak caused by the OTS.

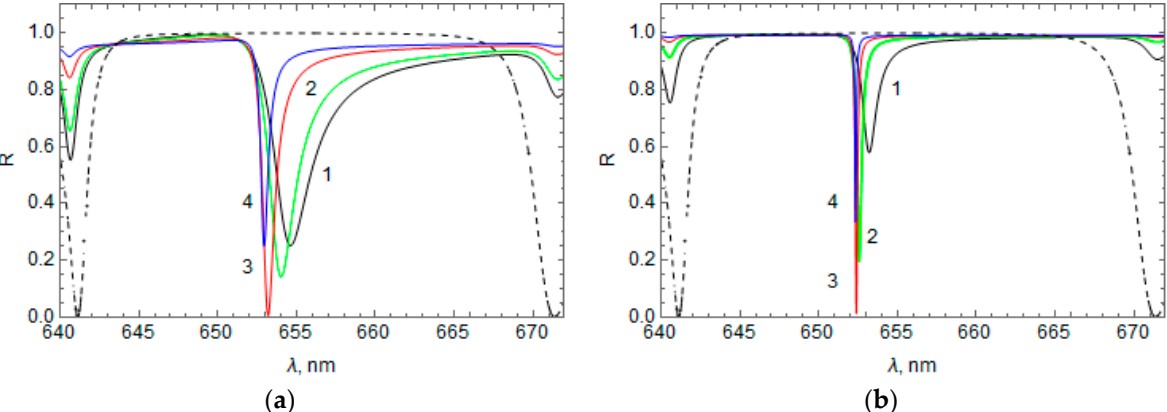

**Figure 5.** Influence of the metal film thickness $l$ on the reflection coefficient of the hybrid structure: (**a**) Au-HPLCG(5CB), $l(\text{nm}) = 25^-(1)$, $30^-(2)$, $45^-(3)$, $60^-(4)$; (**b**) Ag-HPLCG(5CB), $l(\text{nm}) = 30^-(1)$, $45^-(2)$, $60^-(3)$, $75^-(4)$; $L = 70\,\Lambda$, $l_1 = 50\,\text{nm}$, $\bar{\theta}_1 = \pi/2$, $\bar{\theta}_2 = 0$. The dotted line shows the HPLCG band gap.

To understand the dependencies observed in Figures 4 and 5, it should be taken into account that the OTS arises due to the interference of forward and backward light waves, which are repeatedly reflected at the metal film–HPLCG interface. When changing the type of metal and LC, as well as their thicknesses, the coefficient of the internal reflections changes, affecting the conditions and the probability of the OTS occurrence and, consequently, the absorption/reflection coefficients at the OTS wavelength.

The spacer located between the metal film and the HPLCG affects the phases of forward and backward propagating light waves, and, consequently, the conditions for their constructive interference, which ensures the OTS appearance. The difference in the incursion of the phases of these waves caused by the spacer with the thickness $l_1$ and the refractive index $n_1$ is $\varphi(\lambda) = 2\pi \frac{2l_1}{\lambda} n_1$. For a fixed $\varphi = \varphi_0$ corresponding to an interference maximum at a wavelength $\lambda$ associated with OTS, an increase in either $l_1$ or $n_1$ leads to an increase in that OTS wavelength. This agrees with results of the calculations shown in Figure 6a at constant $n_1$ and in Figure 6b at constant $l_1$, as well as with the results obtained for the two-layer HPLCG system described in [39].

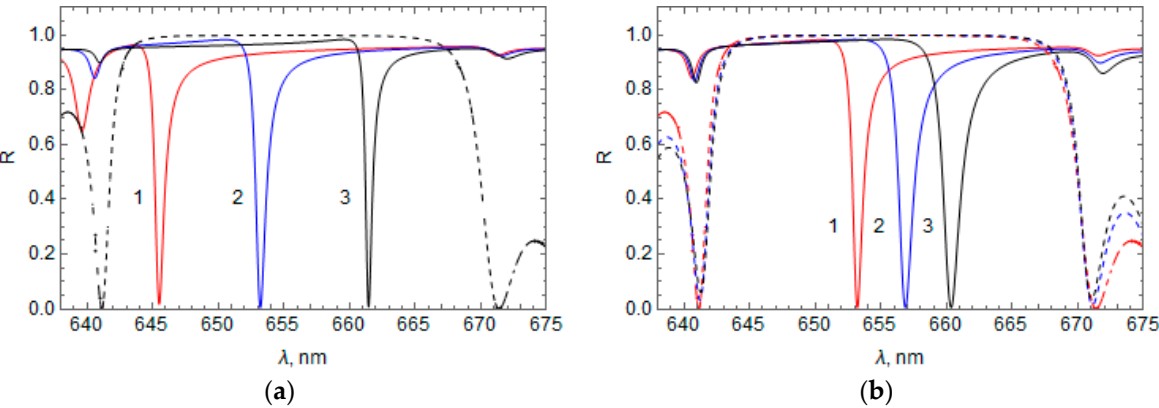

**Figure 6.** Influence of the thickness $l_1$ and the refractive index $n_1$ of the spacer on the reflection coefficient of Au–HPLCG(5CB). (**a**) $n_1 = constant = 1$, $l_1(\text{nm}) = 10^-(1)$, $50^-(2)$, $100^-(3)$; (**b**) $l_1 = constant = 50\,\text{nm}$, $n_1 = 1^-(1)$, $1.5^-(2)$, $1.8^-(3)$; $\bar{\theta}_1 = \pi/2, \bar{\theta}_2 = 0$, $l = 45\,\text{nm}$, $L = 70\,\Lambda$. The dotted lines show the HPLCG band gap.

As in the case of the two-layer HPLCG, an increase in the refractive index $n_p$ and the thickness $d$ of the polymer layers leads to a shift of the OTS dip towards the long-wavelength side. As follows from Equation (9), this shift is due to an increase in the HPLCG average refractive index $\bar{n}$, and in turn the Bragg wavelength $\lambda_r = 2\Lambda\,\bar{n}$, which determines the spectral position of the band gap and, hence, the OTS.

Since the HPLCG average refractive index $\bar{n}$ also depends on the director orientation in the LC layers, the spectral position and magnitude of the reflection dip (as well as the HPLCG band gap) depend on the director angles $\bar{\theta}_1$ and $\bar{\theta}_2$ in the LC1 layer and the LC2 layer, respectively. This dependence is demonstrated in two different cases: (1) when the director angle in LC1, $\bar{\theta}_1$, is fixed and the director angle in LC2, $\bar{\theta}_2$, is varied, and (2) when the director angle $\bar{\theta}_2$ is fixed and the director angle $\bar{\theta}_1$ is varied. Figure 7a,c show the case where $\bar{\theta}_2$ is varied and $\bar{\theta}_1 = \pi/2$ and 0, respectively. Figure 7b,d show the case where $\bar{\theta}_1$ is varied and $\bar{\theta}_2 = \pi/2$ and 0, respectively.

In all cases shown in Figure 7, the reflection dip shifts to the long-wavelength side with a decrease in the director angle $\bar{\theta}_1$ or $\bar{\theta}_2$. The reflection dips from the OTS appear in non-overlapping spectral intervals, namely in the left part of the band gap if the director direction is fixed in the LC1 layers and changes in the LC2 layers, or in right part of the band gap, if the director direction is fixed in the LC2 layers and changes in the LC1 layers. Note that in the two-layer structure [39], applying the same values of the other structure parameters as used with the four-layer structures, the OTS appears in the central part of the HPLCG band gap (which shifts with the OTS as the LC director angle changes). Since the Bragg wavelength in the four-layer structure depends on the director angles in both LC layers, a change in the director orientation in one of the layers affects both the OTS wavelength and the associated reflection dip depth to a lesser extent than in the two-layer structure with the director change by the same angle. Because of this, the spectral response of the four-layer system can be more resistant to director fluctuations due to the temperature instabilities or inhomogeneity of the LC layers.

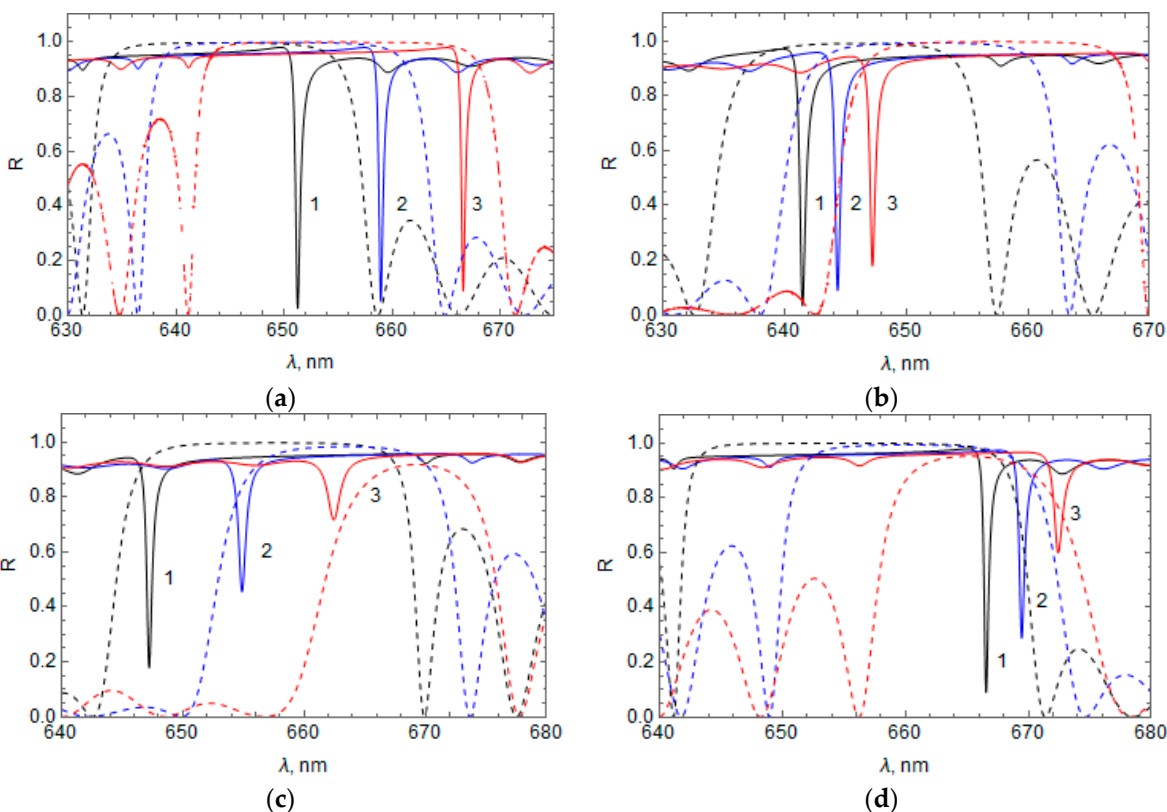

**Figure 7.** Reflectance spectra of the hybrid structure Au–HPLCG(5CB) for different values of the director angle in the LC1 and LC2 layers: **(a)** $\bar{\theta}_1 = \pi/2$, $\bar{\theta}_2 = \pi/4 - (1)$, $\pi/6 - (2)$, $0 - (3)$; **(b)** $\bar{\theta}_2 = \pi/2$, $\bar{\theta}_1 = \pi/4 - (1)$, $\pi/6 - (2)$, $0 - (3)$; **(c)** $\bar{\theta}_1 = 0$, $\bar{\theta}_2 = \pi/2^-(1)$, $\pi/3^-(2)$, $\pi/4^-(3)$; **(d)** $\bar{\theta}_2 = 0$, $\bar{\theta}_1 = \pi/2^-(1)$, $\pi/3^-(2)$, $\pi/4^-(3)$; $l = 45$ nm, $l_1 = 150$ nm, $L = 70\,\Lambda$. The dotted lines show the HPLCG band gap.

The influence of the four-layer HPLCG grating spacing on the reflection spectrum of the hybrid structure Au–HPLCG(5CB) is shown in Figure 8a. It can be seen that with the grating spacing change, the reflection dip shifts together with the HPLCG band gap. Figure 8b illustrates the special case when the director angles in both layers LC1 and LC2 become equal, $\bar{\theta}_1 = \bar{\theta}_2 = 0$. As an example, the four-layer HPLCG with the grating spacing $\Lambda = 0.4\,\mu m$ and $\bar{\theta}_1 = \pi/2$, $\bar{\theta}_2 = 0$ transforms into the two-layer HPLCG with the grating spacing $\Lambda = 0.2\,\mu m$ when $\bar{\theta}_1 = \bar{\theta}_2 = \pi/2$. In this case, the OTS reflection dip in the band gap of the four-layer HPLCG (see plot {A} in Figure 8b) disappears together with the band gap, but a new OTS reflection dip appears in the band gap of the two-layer HPLCG with approximately two times shorter wavelength (plot {B} in Figure 8b). The magnitude of new reflection dip and its spectral position in the two-layer HPLCG band gap depends on the chosen director angle $\bar{\theta}$ as well as on the other parameters of the two-layer HPLCG (see [39]).

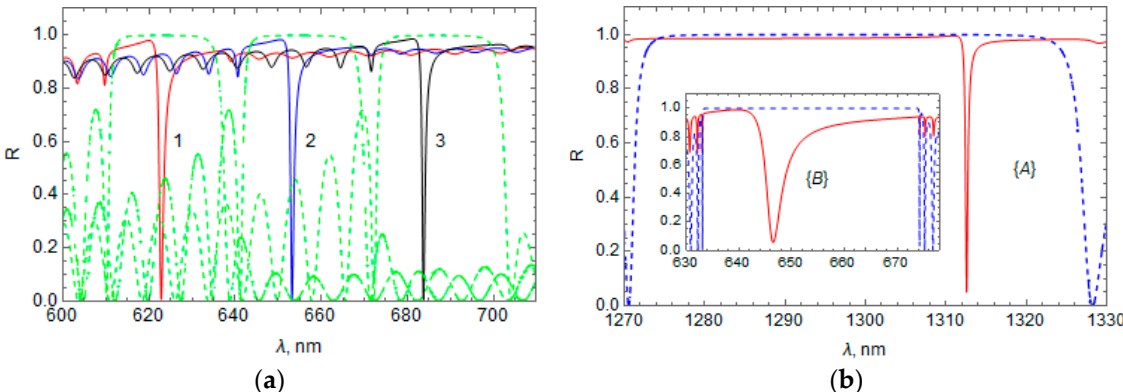

**(a)**                                        **(b)**

**Figure 8.** (**a**) Dependence of the reflection dip and the HPLCG band gap on the grating spacing: $\Lambda\,(\mu m) = 0.19^-(1)$, $0.20^-2$, $0.21^-3$; $l = 45\,nm$, $l_1 = 50\,nm$, $L = 70\,\Lambda$, $\bar\theta_1 = \pi/2$, $\bar\theta_2 = 0$. (**b**) Change in the reflection spectrum if the director angles in layers LC1 and LC2 become the same: {A} $\bar\theta_1 = \pi/2$, $\bar\theta_2 = 0$, $\Lambda = 0.4\,\mu m$; {B} $\bar\theta_1 = \bar\theta_2 = \pi/2$, $\Lambda = 0.2\,\mu m$, $l = 35\,nm$, $l_1 = 270\,nm$, $L = 70\,\Lambda$ The dotted lines show the HPLCG band gap.

## 4. Conclusions

A hybrid structure formed by a metal film and an HPLCG consisting of the four-layer cells can be used for OTS excitation and tuning. Each such cell is a sequence of polymer–LC1–polymer–LC2 layers. LC1 and LC2 differ only in the director orientation, so a hybrid structure with a four-layer HPLCG can be obtained from a hybrid structure with a two-layer HPLCG by applying different voltages to the LC layers of the neighboring cells. Analytical expressions for the reflection, transmission, and absorption coefficients of the hybrid structure in the four-layer HPLCG band gap region are obtained. Numerical calculations show the appearance of the reflection dip and transmission/absorption peaks in the spectral region of the HPLCG band gap due to the excitation of the OTS. The transmission peak disappears with a sufficiently thick HPLCG, and the dip in reflection is entirely due to the energy loss of the incident light wave to the OTS excitation.

The spectral position and magnitude of the reflection dip depend on the type of metal film and LC, the refractive indices and the thicknesses of all layers of the hybrid structure, as well as the director orientation in both LC layers of the HPLCG cells. These parameters affect the coefficient of the internal reflection at the metal–HPLCG interface, thereby affecting the conditions and probability of the OTS occurrence and, consequently, the OTS spectral manifestation. Unlike the hybrid structure with the two-layer HPLCG, there is an optimal metal film thickness corresponding to the largest reflection dip. In addition, the OTS in the structure with the four-layer HPLCG manifests itself in other parts of the band gap compared to the case of the structure with the two-layer HPLCG. This gives more opportunities to influence the position and depth of the OTS reflection dip. A change in the director orientation in each of the LC layers of the four-layer HPLCG affects the OTS wavelength and the reflection dip depth to a lesser extent than in the case of the two-layer HPLCG for the director change by the same angle; therefore, the spectral response of the four-layer structure can be more resistant to the director fluctuations due to the temperature instabilities or inhomogeneity of the LC layers. It also becomes possible to switch from OTS in the band gap of the four-layer HPLCG to OTS in the band gap of the two-layer HPLCG and vice versa.

**Author Contributions:** Conceptualization, V.Y.R.; formal analysis, T.J.B.; investigation, I.P.P. and V.Y.R.; writing—original draft preparation, I.P.P.; writing—review and editing, D.R.E. and T.J.B.; supervision, D.R.E. All authors have read and agreed to the published version of the manuscript.

**Funding:** This research received no external funding.

**Institutional Review Board Statement:** Not applicable.

**Informed Consent Statement:** Not applicable.

**Data Availability Statement:** Data are contained within the article.

**Acknowledgments:** The authors are grateful to MariaCristina Rumi (AFRL) for helpful discussions, and V.Y.R. and I.P.P. thank Azimuth Corporation (USA) for the support received.

**Conflicts of Interest:** The authors declare no conflicts of interest.

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
