# Peer review of "Excitation and Tuning of Optical Tamm States in a Hybrid Structure with a Metal Film Adjacent to a Four-Layer Polymer–Liquid Crystal Stack"

_photonics, doi:10.3390/photonics11030278_

Round 1

Reviewer 1 Report

Comments and Suggestions for Authors

In this paper, the authors theoretically studied the absorption, reflection, and transmission coefficients of a mixed structure formed by metal film and holographic polymer liquid crystal grating (HPLCG) in the band gap spectral region of HPLCG. HPLCG cells consist of four alternating layers, two layers of polymer, and two layers of the same liquid crystal (LC) but with different orientations of the LC director. I believe that publication of the manuscript may be considered only after the following issues have been resolved.

1.    The curve in Figure 3b suggests that the author adjust the parameters and recalculate, as this curve should not converge.

2.    The curve overlap in Figure 5b is quite obvious, and it is not very clear which curve belongs to which one.

3.    In Figure 2a, the authors mentioned the simulation calculation of COMSOL. Can the authors provide specific calculation parameters?

4.    In the introduction section, regarding Tamm plasmon based devices, the author needs to mention some of the latest related work, such as, Optics & Laser Technology 169, 2024, 110186; Opto-Electron Sci 1, 210010 (2022); Opto-Electron Adv 5, 200098 (2022); Diamond and Related Materials, 136, 2023, 109960.

5.    The English expression of the whole article needs to be further improved.

Comments on the Quality of English Language

Minor editing of English language required.

Reviewer 2 Report

Comments and Suggestions for Authors

In the paper entitled "Excitation and tuning of optical Tamm states in a hybrid structure with a metal film adjacent to a four-layer polymer-liquid crystal stack," the authors calculate the spectral features of optical Tamn states (OTS) at the metal film-holographic polymer-liquid crystal grating (HPLCG) interface. The HPLCG adopted in this work consists of four alternating layers, two layers of polymer, and two layers of the same liquid crystal (LC) but with controllable orientations of the LC director. Compared to the previous literature, in which a similar structure with only two alternating layers of LC and polymer is employed, the four-alternating-layer structure demonstrated in this work shows superior spectral manipulations of OTS.

 I would like to recommend this work for publication in Photonics after some minor issues are adressed.

1. The "reflection coefficient" and "reflectance" have to be distinguished clearly. The "reflection coefficient" is usually defined as the ratio between the amplitudes of the reflected and the incident electric fields, while the "reflectance" is defined as the ratio between the intensities of the reflected and the incident light. Since the light intensity is proportional to the square of the field amplitude, the "R" shown in Eq. (2) should be "reflectance" rather than "reflection coefficient". The definitions of "transmission coefficient" and "absorption coefficient" should also follow the similar rule.

2. Some typos, missing items, formats and fonts  should be checked carefully, for example, the right parenthesis in Eq. (6) is lost, the notation "delta" in Eq. (9) is not defined, the words "withthe" at line 195 should be corrected as "with the", and the fonts in the figure caption of Fig. 6 are different from the other parts in this manuscript. 

Reviewer 3 Report

Comments and Suggestions for Authors

The reviewed article concerns a theoretical analysis of excitation and tuning of optical Tamm states in a hybrid structure with a metal film adjacent to a four-layer polymer-liquid crystal stack. The scheme is interesting and the manuscript has been carefully thought out. Nevertheless, there are several issues requiring additional discussion:

1.     In the introduction, the authors mention selected properties of hybrid photonic structures. It is worth expanding this context a bit to include the latest concepts, e.g.: Advanced Optical Materials (2023): 2302483; Light: Science & Applications 11.1 (2022): 141.

2.     The authors do not provide source articles from which the physical formulas were taken.

3.     The description of numerical simulations is incomplete. In what program were they made? Was it a commercial program or homemade code? Based on what method did the algorithm work, i.e. TMM, FDTD, etc.? What was the numerical grid and what were the boundary conditions? The description should enable reproduction of the results.

4.     In the discussion, the authors should pay attention to what advantages or other special features the presented model has compared to other (previously published) models? What parameters make it stand out?

Round 2

Reviewer 1 Report

Comments and Suggestions for Authors

 Accept in present form